# Taspase1 Facilitates Topoisomerase IIβ-Mediated DNA Double-Strand Breaks Driving Estrogen-Induced Transcription

**DOI:** 10.3390/cells12030363

**Published:** 2023-01-18

**Authors:** Lisa Oelschläger, Paul Stahl, Farnusch Kaschani, Roland H. Stauber, Shirley K. Knauer, Astrid Hensel

**Affiliations:** 1Department of Molecular Biology II, Center of Medical Biotechnology (ZMB), University Duisburg-Essen, 45141 Essen, Germany; 2Analytics Core Facility Essen (ACE), Center of Medical Biotechnology (ZMB), University Duisburg-Essen, 45141 Essen, Germany; 3Department of Otorhinolaryngology, Molecular and Cellular Oncology, University Mainz Medical Center (UMM), 55131 Mainz, Germany

**Keywords:** stimulus-triggered transcription, hormones, histone epigenetic labelling, H3K4me3, Mixed-lineage leukemia

## Abstract

The human protease Taspase1 plays a pivotal role in developmental processes and cancerous diseases by processing critical regulators, such as the leukemia proto-oncoprotein MLL. Despite almost two decades of intense research, Taspase1’s biology is, however, still poorly understood, and so far its cellular function was not assigned to a superordinate biological pathway or a specific signaling cascade. Our data, gained by methods such as co-immunoprecipitation, LC-MS/MS and Topoisomerase II DNA cleavage assays, now functionally link Taspase1 and hormone-induced, Topoisomerase IIβ-mediated transient DNA double-strand breaks, leading to active transcription. The specific interaction with Topoisomerase IIα enhances the formation of DNA double-strand breaks that are a key prerequisite for stimulus-driven gene transcription. Moreover, Taspase1 alters the H3K4 epigenetic signature upon estrogen-stimulation by cleaving the chromatin-modifying enzyme MLL. As estrogen-driven transcription and MLL-derived epigenetic labelling are reduced upon Taspase1 siRNA-mediated knockdown, we finally characterize Taspase1 as a multifunctional co-activator of estrogen-stimulated transcription.

## 1. Introduction

Human Threonine-Aspartase1 (Taspase1) is translated as a proenzyme of 420 amino acids, which belongs to the type II asparaginases [1,2]. In accordance with other members of this family, Taspase1 is activated by autocatalytic cis-cleavage between D233 and T234 resulting in the two subunits α28 and β22, which are thought to reassemble into a αββα heterodimer [2,3,4,5]. The N-terminal threonine of mature β-subunit acts as a nucleophile enabling Taspase1 to cleave substrates in *trans* after an aspartate residue embedded in a conserved peptide motif [1,2,4,5]. Verified targets comprise the MLL-Proteins (MLL1, MLL2, MLL4), the transcription factor IIa (TFIIA) [6], the upstream stimulatory factor II (USF2) [1], the catalytic subunit of DNA polymerase (REV3L) [7] and Myosin1f [8]. Taspase1 founded a new class of endopeptidases [2], particularly because its cleavage activity is not inhibited by conventional protease-inhibitors [2,9]. Upregulation of Taspase1 expression was detected in several leukemias and also solid tumor types [10,11,12]. Moreover, Taspase1 knockout mice exhibit homeotic transformations of the axial skeleton, a higher postnatal lethality and a smaller size [12], thus underlining Taspase1′s role in developmental processes. Furthermore, Taspase1 was functionally linked to cell cycle progression as Taspase1^−/−^ mice showed a decrease in cyclin levels and elevated amounts of CDK inhibitors finally resulting in a reduced body size [11,12].

However, so far there is no common denominator regarding Taspase1′s verified target proteins that could link the overall function of this protease to a certain cellular process, a superordinate biological pathway or a specific signaling cascade.

To interconnect these fragmented pieces of knowledge and obtain a clearer picture of Taspase1′s sphere of activity, we took a step back and looked coherently at its best characterized substrate, the MLL1 (mixed lineage or myeloid/lymphoid leukemia) protein. MLL is a human H3K4-specific histone methyltransferases and as such plays an important role as an epigenetic regulator of transcription in embryonic development and hematopoiesis [13,14]. For example, it regulates maintenance of *HOX* gene expression through direct promoter binding and histone H3 lysine 4 (H3K4) trimethylation [13,15,16].

Taspase1 is the only protease known to cleave the MLL1 protein at two conserved sites [2]. The resulting subunits heterodimerize and form the scaffold for a multiprotein complex that contains the core components Ash2, Wdr5, Rbbp5, HCF1 and Dpy30 [17,18]. In addition to these common subunits the protein complex has been shown to interact with further facultative components, such as nuclear hormone receptors [16,19,20]. Each MLL protein harbors at least one LXXLL steroid receptor interaction domain, also known as NR box, which mediates the interaction with nuclear receptors (NR) [21,22].

Nuclear receptors are ligand-dependent transcription factors which are activated by hormones or other cell membrane crossing signaling molecules such as retinoic acid and vitamin D [23,24]. The steroid receptor subgroup of the nuclear receptor family includes the estrogen receptor (ER), androgen receptor, progesterone receptor, mineralocorticoid receptor and the glucocorticoid receptor. Upon activation by the corresponding hormone, these receptors bind hormone response elements (HRE) within certain gene promoters [25], thereby regulating the transcription of many genes involved in development, cell proliferation, metabolism and reproduction.

Another essential player in the transcriptional response to hormone stimuli as well as in the generation of harmful leukemic gene translocations is Topoisomerase II (Topo II). Formerly, Type II topoisomerases were mainly known as torsional stress-resolving enzymes that reduce DNA supercoiling, tangles, knots and catenanes [26,27,28,29]. For that purpose they cut both strands of the DNA helix in an ATP-dependent manner [30]. Only vertebrates possess two isoforms of Topo II, namely Topo IIα and Topo IIβ, which show extensive sequence similarity within their N-terminal ATPase and central catalytic domains but diverge substantially in their C-terminal domains [31,32,33,34,35]. Type IIα Topoisomerase is essential for proliferating cells where it is upregulated in G2/M-phase of the cell cycle which is in line with its unraveled functions during replication and sister chromatid decatenation [29,36,37,38,39,40].

Topo IIβ is more ubiquitously expressed in a proliferation independent manner, but its function was less understood and mainly assigned to the resolving of topological constraints for smooth RNA Pol II-related transcription [31,41].

However, in 2006 a novel biological function could be attributed to Topo IIβ: it was revealed that this isomerase is responsible for estrogen-induced transient DNA double-strand breaks (DSBs) within estradiol-responsive promoters, which foster transcriptional initiation [42]. Moreover, it was reported that further physiological stimuli (androgens, retinoic acid, glucocorticoids, insulin and serum) induce Topo IIβ-mediated DSBs in the regulatory regions of responsive genes [43,44,45,46,47,48]. Furthermore, such hormone-provoked DNA-breakage allows stalled RNA Pol II to escape from its pause site and perform transcriptional elongation [46]. There is also emerging evidence that those programmed DNA breaks in hormone-induced transcription are accompanied by activation of DNA damage repair pathways. Topo IIβ, which is recruited to the promoters of hormone-responsive genes, forms a large complex containing DNA-PKcs, Ku, and PARP-1 [42].

Following this line of reasoning, we could now hypothesize a higher level-function of Taspase1 in estrogen-driven transcription as this protease affects diverse players acting in the response to hormone stimuli: it enhances Topo II-mediated DNA cleavage and alters the epigenetic signature upon 17β-estradiol stimulus by cleavage of chromatin modifying enzyme MLL.

## 2. Materials and Methods

### 2.1. Cell Culture and Reagents

The following cells were used in the analysis: HEK-293T (ATCC CRL-3216, LGC Standards, Wesel, Germany), adherent culture; HeLa Kyoto (from S. Narumiya, Kyoto University, Kyoto, Japan) and MCF-7 (ATCC HTB-22, LGC Standards, Wesel, Germany), adherent culture. Cells lines were regularly checked for mycoplasma contamination and all experiments were performed with mycoplasma-free cells. Cells were cultivated at 37 °C in a relative humidity of about 95% and 5% CO_2_ and were maintained in Dulbecco’s Modified Eagle’s Medium (DMEM, Thermo Fisher Scientific, Dreieich, Germany) supplemented with 10% (*v*/*v*) fetal calf serum (FCS, Thermo Fisher Scientific, Dreieich, Germany) and 1% (*v*/*v*) Gibco^®^ antibiotic-antimycotic (Thermo Fisher Scientific, Dreieich, Germany). At a confluency of about 70 to 90%, the cells were passaged in an appropriate ratio of 1:10 to 1:20. If not stated otherwise, standard chemicals were purchased from AppliChem, Darmstadt, Germany.

### 2.2. Lovastatin-Mediated G1 Arrest

A 10 mM stock solution was prepared by dissolving lovastatin (Sigma-Aldrich, Taufkirchen, Germany) in 70% Ethanol. Asynchronous cultures of HEK 293T cells were seeded in culture dishes with 60 mm diameter at a cell density of 0.6 × 10^5^ cells/cm^2^ in 3 mL DMEM. After approx. 24 h, cells were treated with 20 µM lovastatin. After 24 h, synchronized cells were subjected to 17β-estradiol treatment in lovastatin-containing medium.

### 2.3. 17β-estradiol Treatment and RIPA Cell Lysis

A total of 24 h after seeding HEK 293T- or MCF7 cells in 3 mL culture medium in 60-mm-diameter cell culture dishes, the medium was replaced by 3 mL hormone-reduced medium (Opti-MEM supplemented with 10% (*v*/*v*) charcoal-stripped FBS (Thermo Fisher Scientific, Dreieich, Germany) and 1% (*v*/*v*) antibiotic-antimycotic (Gibco) for up to 4 h. Subsequently, cells were treated with 50 nM 17β-estradiol (Sigma-Aldrich, Taufkirchen, Germany) for indicated time periods and immediately afterwards lysed directly in the dish in RIPA lysis buffer (50 mM Tris-HCl pH 7.4, 150 mM NaCl, 5 mM EDTA, 1% NP-40, 1% Natriumdeoxycholat, 1 mM DTT, 1 mM PMSF and a complete protease inhibitor cocktail tablet from Roche Diagnostics, Mannheim, Germany). In this respect it should be noted that common protease inhibitors are not able to block Taspase1 cleavage activity [2]. RIPA cell lysates were scraped into Eppendorf tubes (Eppendorf, Wesseling, Germany) and were sonicated twice (30 s at 90% amplitude) with a Sonopuls mini20 device (Bandelin, Berlin, Germany) at 4 °C. Cell debris were removed (14,000 rpm, 20 min, 4 °C) and protein concentrations were determined and adjusted using the Bradford Reagent (Protein Assay Dye Reagent Concentrate) from Bio-Rad Laboratories, Feldkirchen, Germany. The α-Tubulin was used to control the equal loading of lysates.

### 2.4. Transfection

For co-immunoprecipitation experiments, HEK 293T cells were seeded in standard TC dish 100 (Sarstedt, Nümbrecht, Germany) with 10 mL DMEM complemented with 10% (*v*/*v*) FCS and 1% (*v*/*v*) antibiotic-antimycotic. After 24 h, the cells were transiently transfected with a plasmid coding for wild type (WT) Taspase1 and a catalytic inactive Taspase1 variant (Taspase1^T234V^) tagged with GFP or HA or GFP-tagged Topoisomerase IIα and Topoisomerase IIβ (both provided from [49]) using calcium phosphate.

For siRNA experiments, HEK 293T- and HeLa-cells were seeded in a standard TC dish 60 (Sarstedt, Nümbrecht, Germany) with 3 mL DEMEM supplemented with 10% (*v*/*v*) FCS and 1% (*v*/*v*) antibiotic-antimycotic. After 24 h, the cells were transfected with TASP1 siRNA (Sigma-Aldrich, Taufkirchen, Germany, SASI_Hs01_00187376 and SASI_Hs01_00187378) using Lipofectamine^®^ RNAiMAX Reagent (Thermo Fisher Scientific, Dreieich, Germany) according to the manufacturer’s protocol. The siRNA single strands were pooled and annealed according to manufacturer’s protocol before use. As control, cells were transfected with a non-targeting siRNA negative universal control (Invitrogen 46-2001, Thermo Fisher Scientific, Dreieich, Germany).

### 2.5. Immunoprecipitation

The 293T cells were harvested 24 h after transfection in a low-salt lysis buffer (50 mM Tris pH 8.0, 150 mM NaCl, 5 mM EDTA, 0.5% NP-40, 1 mM DTT, 1 mM PMSF and a complete protease inhibitor cocktail tablet (Roche Diagnostics, Mannheim, Germany) at 4 °C. After at least three sonification steps (20 s at 90% amplitude) with a Sonopuls mini20 device (Bandelin, Berlin, Germany), cell debris was removed (14,000 rpm, 20 min, 4 °C) to obtain whole cell lysates for immunoprecipitation. Alternatively, chromatin extract fractions were prepared from 293T cells with the Chromatin Extraction Kit (ab117152, Abcam, Cambridge, UK) according to the manufacturer’s instructions.

Co-immunoprecipitations were performed using μMACS^TM^ GFP-tagged Protein Isolation Kit and µ-MACS (magnetic antibody cell sorting) columns (Miltenyi Biotec, Bergisch Gladbach, Germany) according to the suppliers’ recommendation. Briefly, whole cell lysates or chromatin fractions were incubated with 50 µL µ-MACS anti-GFP MicroBeads for 30 min on ice. The μMACS^TM^ GFP tagged Protein Isolation Kit was used according to manufacturer’s protocol except that the washing steps were changed as follows: the columns were washed once with 200 µL wash buffer-1 (Miltenyi Biotec, Bergisch Gladbach, Germany), three times with 200 µL low salt lysis buffer and once with 100 µL wash buffer-2 (Miltenyi Biotec, Bergisch Gladbach, Germany). Afterwards, pre-heated elution buffer (Miltenyi Biotec, Bergisch Gladbach, Germany) was applied onto each column according to manufacturer’s recommendation. For mass spectrometry sample preparation, a self-prepared elution buffer (50 mM Tris pH 6.8, 1 mM EDTA, 1% SDS and 50 mM DTT) was utilized. The eluates and the input samples of the lysates or chromatin extract were subjected to SDS-PAGE and immunoblotting.

### 2.6. SDS-PAGE and Immunoblotting

Immunoprecipitates, chromatin extracts, total protein lysates and RIPA cell lysates samples were separated by SDS-PAGE (7.5 to 10% acrylamide) and subsequently transferred to polyvinylidene difluoride (PVDF) membranes (Amersham Hybond, GE Healthcare Europe, Solingen, Germany) with a PerfectBlue™ tank electro blotter (Peqlab Biotechnologie, Erlangen, Germany). Membranes were probed with the following primary antibodies: anti-c-fos (ab134122, Abcam, Cambridge, UK), anti-c-myc (Invitrogen 13-2500, Thermo Fisher Scientific, Dreieich, Germany), anti-Cyclin E (32-1600, Thermo Fisher Scientific, Dreieich, Germany), anti-ERα (NBP1-84827, Novus Biologicals, Wiesbaden, Germany), anti-GFP (sc-9996, Santa Cruz Biotechnology, Heidelberg, Germany), anti-HA (ab9110, Abcam, Cambridge, UK), anti-Histone H3 K4 trimethylation (9751, Cell signaling Technology, Leiden, The Netherlands), anti-HOXA9 (ab140631, Abcam, Cambridge, UK), anti-MLL/KMT2A (NB600-248, Novus Biologicals, Wiesbaden, Germany), anti-Nucleolin (ab22758, Abcam, Cambridge, UK), anti-Nucleophosmin1/B23 (sc-47725, Santa Cruz Biotechnology, Heidelberg, Germany), anti-Topoisomerase IIα/β (ab109524, Abcam, Cambridge, UK), anti-Topoisomerase IIα (sc-365071, Santa Cruz Biotechnology, Heidelberg, Germany), anti-Topoisomerase IIβ (sc-365916, Santa Cruz Biotechnology, Heidelberg, Germany), anti-Taspase1 (AP11325PU-N, OriGene Technologies, Herford, Germany), anti-α-Tubulin (T6074, Sigma-Aldrich, Taufkirchen, Germany) or anti-γH2AX (613402, BioLegend, Amsterdam, The Netherlands).

For visualization of immune complexes, HRP-conjugated secondary antibodies were applied (NXA931 and NA934, GE Healthcare Europe, Solingen, Germany), and detected with the Pierce^TM^ ECL Plus Western Blotting Substrate or SuperSignal^TM^ West Femto Maximum Sensitivity Substrate from Thermo Fisher Scientific, Dreieich, Germany and the ChemiDoc MP Imaging System (Bio-Rad Laboratories, Feldkirchen, Germany).

### 2.7. Proximity Ligation Assay

All washing steps were performed with PBS. HeLa cells were seeded in 35 mm glass bottom dishes (MatTek Corporation, Ashland, MA, USA) and after 24 h, the cells were washed and fixed with Roti^®^-Histofix 4% (Carl Roth, Karlsruhe, Germany) for 20 min at RT. Following three washing steps, the cells were permeabilized, and unspecific binding sites were blocked with 0.3% Triton X 100, 5% goat serum in DPBS (blocking buffer) for 30 min at RT. Afterwards, the cells were incubated at 4 °C overnight with the two primary antibodies rabbit-anti-Topoisomerase IIα/β (ab109524, Abcam, Cambridge, UK) and mouse-anti-Taspase1 (sc-390934, Santa Cruz Biotechnology, Heidelberg, Germany) that were diluted in 0.3% Triton-X-100, 1% BSA in DPBS (antibody dilution buffer). The PLA staining was performed with the Duolink^®^ In-Situ-Detection Reagents Orange, Duolink^®^ In Situ PLA^®^ Probe Anti-Mouse MINUS and Anti-Rabbit PLUS (Sigma-Aldrich, Taufkirchen, Germany) according to manufacturer’s Duolink^®^ PLA Fluorescence Protocol. Staining of cell nuclei and membranes was attained with 10 μg/mL Hoechst33342 and 1:5000 diluted HCS CellMask™ Deep Red Stain (Thermo Fisher Scientific, Dreieich, Germany) in PBS for 30 min at RT. Cells were stored at 4 °C in 0.1% (*w*/*v*) sodium azide in PBS until they were analyzed by fluorescence microscopy. Maximum projection images of z-stacks were taken with the confocal laser scanning microscope SP8X Falcon (Leica Microsystems, Wetzlar, Germany) using the Leica Application Suite X (LAS X 3.5.7) software (Leica Microsystems, Wetzlar, Germany) and the HC PL APO 63x/1.2 W motCORR CS2 water-immersion objective. This microscope is equipped with different lasers (Diode: 405 nm; Argon, 458/476/488/496/514 nm; WLL E: 470–670 nm; UVA: 355 nm) and various detectors (PMT confocal imaging detectors and sensitive imaging hybrid detector). PLA foci were analyzed with the cell image analysis software Cell Profiler 4.1.3 (Broad Institute, Cambridge, MA, USA) and assigned to the nuclear and the whole cell masks accordingly.

### 2.8. Protein Expression and Purification

The pET22b-WT Taspase1-His was expressed in *E. coli* BL21-CodonPlus (DE3)-RIL. Protein expression was induced at an OD600 of 0.6–0.8 by adding IPTG to a final concentration of 0.2 mM and continued for 6 h at 30 °C and 120 rpm. Bacterial cells were harvested, lysed with lysozyme and subsequently sonicated in wash buffer (50 mM Na_2_HPO_4_, 450 mM NaCl, 10 mM imidazole, pH 8.0) supplemented with 1 mM PMSF. The protein was purified on a HisTrap FF column (GE Healthcare Europe, Solingen, Germany) eluted with a linear gradient up to 250 mM imidazole in 50 mM Na_2_HPO_4_, pH 8.0, containing 450 mM NaCl. Taspase1-His-containing fractions were pooled, and a size exclusion chromatography was performed with a HiLoad16/600 Superdex 200 pg column (GE Healthcare Europe, Solingen, Germany) in 50 mM Na_2_HPO_4_, pH 8.0 with 450 mM NaCl and 1 mM DTT. The pure protein was concentrated using VivaSpin Turbo with a molecular weight cutoff of 30 kDa (Sarotius Stedim Biotech, Göttingen, Germany), subsequently shock-frozen in liquid nitrogen and stored at −80 °C.

### 2.9. Electrophoretic Mobility Shift Assay (EMSA)

A reaction mix of 10 μL was prepared with 0.5–4.0 µM purified recombinant Taspase1-His and 0.5 μg of the plasmid pBR322 (Thermo Fisher Scientific, Dreieich, Germany), primarily consisting of supercoiled DNA. The reaction mix was incubated for 15 min at 37 °C and 300 rpm. The samples were immediately mixed with 6 × TriTrack DNA Loading Dye (Thermo Fisher Scientific, Dreieich, Germany) to stop the reaction and loaded on a 0.7% agarose gel supplemented with HD Green Plus DNA Stain (INTAS Science Imaging Instruments, Göttingen, Germany). Size separation occurred at a constant voltage of 90 V with the power supplier PeqPower 300 (Peqlab Biotechnologie, Erlangen, Germany) for 60 min using 1 × Tris-Acetate EDTA as running buffer. Separated protein-DNA complexes were visualized with the ChemiDoc MP Imaging System (Bio-Rad Laboratories, Feldkirchen, Germany).

### 2.10. Topoisomerase II DNA Cleavage Assay

The Topoisomerase II DNA cleavage assay was performed as described [50]. The reaction mix contained 4 units of recombinant Topoisomerase IIβ (Inspiralis Limited, Norwich, UK), 0.1–4.0 µM purified recombinant Taspase1-His and 1 µg pBR322 in 50 mM Tris-HCl pH 7.5, 125 mM NaCl, 10 mM MgCl_2_, 5 mM DTT and 100 µg/mL albumin. After incubation at 37 °C and 300 rpm for 20 min, protein degradation of Topoisomerase IIβ and Taspase1-His was attained by the addition of Proteinase K (Roche Diagnostics, Mannheim, Germany) to a final concentration of 0.45 mg/mL for 30 min at 45 °C and 300 rpm. Subsequently, the samples were separated by electrophoresis using 1 × Tris-Acetate EDTA and a 0.7% agarose gel supplemented with HD Green Plus DNA Stain (INTAS Science Imaging Instruments, Göttingen, Germany) at 90 V for 60 min. DNA was visualized by the ChemiDoc MP Imaging System (Bio-Rad Laboratories, Feldkirchen, Germany).

### 2.11. In-Solution Tryptic Digestion

After cell lysis, proteins were taken up in digestion buffer (6 M Urea, 0.2% RapiGest SF Surfacatant (Waters, Eschborn, Germany) in 100 mM Ammonium bicarbonate (ABC, Sigma-Aldrich, Taufkirchen, Germany) containing RapiGest SF Surfacatant (used as in the manufacturers guidelines). Samples were subsequently incubated with DTT (Dithiothreitol) at a final concentration of 5 mM on a ThermoMixer C (Eppendorf, Wesseling, Germany) for 30 min at 1300 rpm and RT. This was followed by an incubation with IAM (Iodoacetamide) at a final concentration of 20 mM for 30 min at 1300 rpm and RT in the dark. Excess IAM was then quenched by adding DTT to a final concentration of 25 mM. After reduction and alkylation, digestion was initiated by adding 500 ng LysC (FUJIFILM Wako Chemicals Europe, Neuss, Germany) for 3 h at 1300 rpm and 37 °C. After 3 h, 500 ng Trypsin (Promega Corporation, Walldorf, Germany) were added. Samples were subsequently incubated at 37 °C for 16 h while shaking at 1300 rpm. Digestion was stopped by adding formic acid (FA, Thermo Fisher Scientific, Dreieich, Germany) to a final concentration of 5%. The samples were then incubated for further 45 min at 37 °C to cleave the RapiGest detergent. The white precipitate that was visible after 45 min was pelleted (10 min, 13,000 × g). The supernatant was carefully removed and subsequently desalted on home-made 2 disc C18 StageTips as described [51]. After elution from the StageTips, samples were dried using a vacuum concentrator (Eppendorf, Wesseling, Germany), and the peptides were taken up in 10 µL 0.1% formic acid solution.

### 2.12. LC-MS/MS

Experiments were performed on an Orbitrap Elite instrument (Thermo Fisher Scientific, Dreieich, Germany) [52] that was coupled to an EASY-nLC 1000 liquid chromatography (LC) system (Thermo Fisher Scientific, Dreieich, Germany). The LC was operated in the one-column mode. The analytical column was a fused silica capillary (75 µm × 46 cm) with an integrated PicoFrit emitter (New Objective, Littleton, MA, USA) packed in-house with Reprosil-Pur 120 C18-AQ 1.9 µm resin (Dr. A. Maisch HPLC, Ammerbuch-Entringen, Germany). The analytical column was encased by a column oven (Sonation, Biberach an der Riß, Germany) and attached to a nanospray flex ion source (Thermo Fisher Scientific, Dreieich, Germany). The column oven temperature was adjusted to 45 °C during data acquisition. The LC was equipped with two mobile phases: solvent A (0.1% formic acid, FA, in water) and solvent B (0.1% FA, 20% water and 80% acetonitrile, ACN). All solvents were of UHPLC grade (Honeywell International, Seelze, Germany). Peptides were directly loaded onto the analytical column with a maximum flow rate that would not exceed the set pressure limit of 980 bar (usually around 0.6–1.0 µL/min). Peptides were subsequently separated on the analytical column by running a 140 min gradient of solvent A and solvent B (start with 9% B; gradient 9% to 44% B for 110 min; gradient 44% to 100% B for 20 min and 100% B for 10 min) at a flow rate of 300 nl/min. The mass spectrometer was operated using Xcalibur software (version 2.2 SP1.48, Thermo Fisher Scientific, Dreieich, Germany). The mass spectrometer was set in the positive ion mode. Precursor ion scanning was performed in the Orbitrap analyzer (FTMS; Fourier Transform Mass Spectrometry) in the scan range of *m*/*z* 300–1800 and at a resolution of 60,000 with the internal lock mass option turned on (lock mass was 445.120025 *m*/*z*, polysiloxane) [53]. Product ion spectra were recorded in a data dependent fashion in the ion trap (ITMS) in a variable scan range and at a rapid scan rate. The ionization potential (spray voltage) was set to 1.8 kV. Peptides were analyzed using a repeating cycle consisting of a full precursor ion scan (3.0 × 10^6^ ions or 50 ms) followed by 15 product ion scans (1.0 × 10^4^ ions or 60 ms) where peptides are isolated based on their intensity in the full survey scan (threshold of 500 counts) for tandem mass spectrum (MS2) generation that permits peptide sequencing and identification. Collision induced dissociation (CID) energy was set to 35% for the generation of MS2 spectra. During MS2 data acquisition dynamic ion exclusion was set to 120 s with a maximum list of excluded ions consisting of 500 members and a repeat count of one. Ion injection time prediction, preview mode for the FTMS, monoisotopic precursor selection and charge state screening were enabled. Only charge states higher than 1 were considered for fragmentation.

### 2.13. Peptide and Protein Identification Using MaxQuant

RAW spectra were submitted to an Andromeda [54] search in MaxQuant (1.6.10.43) using the default settings [55]. Label-free quantification and match-between-runs was activated [56]. The MS/MS spectra data were searched against the Uniprot *H. sapiens* reference database (UP000005640_9606.fasta, 74,053 entries, downloaded 11/9/2019) and a dedicated database containing the GFP sequence (ACE_0496_GFP.fasta; 1 entry). All searches included a contaminants database search (as implemented in MaxQuant, 245 entries). The contaminants database contains known MS contaminants and was included to estimate the level of contamination. Andromeda searches allowed oxidation of methionine residues (16 Da) and acetylation of the protein N-terminus (42 Da) as dynamic modifications and the static modification of cysteine (57 Da, alkylation with iodoacetamide). Enzyme specificity was set to “Trypsin/P” with two missed cleavages allowed. The instrument type in Andromeda searches was set to Orbitrap and the precursor mass tolerance was set to ±20 ppm (first search) and ± 4.5 ppm (main search). The MS/MS match tolerance was set to ±0.5 Da. The peptide spectrum match FDR and the protein FDR were set to 0.01 (based on target-decoy approach). Minimum peptide length was 7 aa. For protein quantification unique and razor peptides were allowed. Modified peptides were allowed for quantification. The minimum score for modified peptides was 40. Label-free protein quantification was switched on, and unique and razor peptides were considered for quantification with a minimum ratio count of 2. Retention times were recalibrated based on the built-in nonlinear time-rescaling algorithm. MS/MS identifications were transferred between LC-MS/MS runs with the “match between runs” option in which the maximal match time window was set to 0.7 min and the alignment time window set to 20 min. The quantification is based on the “value at maximum” of the extracted ion current. At least two quantitation events were required for a quantifiable protein. Further analysis and filtering of the results was completed in Perseus v1.5.5.3. [57]. For quantification we combined related biological replicates to categorical groups and investigated only those proteins that were found in at least one categorical group in a minimum of three out of four biological replicates. Comparison of protein group quantities (relative quantification) between different MS runs is based solely on the LFQ’s as calculated by MaxQuant, MaxLFQ algorithm [56].

### 2.14. Statistical Analysis

For experiments stating *p*-values one-way ANOVA followed by Tukey’s multiple comparison test was performed. The *p*-values represent data obtained from five or more independent experiments with at least 150 (and up to 450) measurements (cells). The *p*-values < 0.05 were considered as significant.

## 3. Results

### 3.1. Taspase1 Interacts with Diverse Players of the Cellular Estrogen Response

Some of Taspase1‘s verified target proteins such as USF2 [1] and TFIIA [6,58] have been annotated as estrogen-regulated genes, as well as the known Taspase1 interaction partner Nucleophosmin1 (NPM1). MLL proteins, the most prominent Taspase1 substrates, interact with hormone receptors (NR) and are involved in steroid-hormone induced transcriptional activation [22,59]. For a more comprehensive understanding of Taspase1′s sphere of activity, we elucidated whether further players in the cellular estrogen response might likewise be able to interact with Taspase1. For that purpose, protein complexes were isolated from asynchronous HEK 293T cells using ectopically expressed Taspase1-GFP. After co-immunoprecipitation (co-IP) with anti-GFP Ab-coated magnetic beads (Figure 1A), the isolated wild type (WT) Taspase1-GFP and the cleavage-incompetent mutant T234V were screened for potential binding partners using diverse antibodies. The known interaction partner NPM1 was co-immunoprecipitated in significant amounts with Taspase1-GFP (Figure 1B) but to a lesser extent with the inactive mutant concordant with earlier studies [10]. Co-IP experiments identified some novel proteins associated with Taspase1-containing complexes: Nucleolin, Topo IIα/β and γH2AX could be co-isolated with WT Taspase1-GFP and the Taspase1^T234V^-GFP cleavage mutant, but not with GFP alone. Additionally, the composition of immunoprecipitated Taspase1-GFP complexes was investigated by tandem mass spectrometry, where the newly identified interactors Topo IIα, Topo IIβ and Nucleolin could be verified (Figure 1C). In addition, MS-data analysis revealed approximately 300 further proteins associated with isolated Taspase1-GFP (Appendix A), e.g., general transcription factors required for RNA polymerase III-mediated transcription, diverse nucleolar proteins, mediator complex proteins and ribosomal proteins.

### 3.2. Taspase1 Is Estrogen-Inducible and Involved in Estrogen-Driven Transcription

To assess if the expression of Taspase1 and its newly identified interactors was estradiol-inducible, RIPA cell lysates of 293T cells were prepared at defined time points after 17β-estradiol (E2) treatment (50 nM) and analyzed for differential protein expression. 

Indeed, Taspase1, Topo IIα/β, histone H3 lysine 4 trimethylation (H3K4me3) and HOXA9 (as a MLL target gene) revealed a cyclic upregulation in response to estrogen stimulation, with a first peak within 20–40 min in 293T cells (Figure 2A and corresponding densitometric analyses in Appendix A) and within 40 min in MCF-7 cells (Appendix A) as well as a second peak around 2 h. This expression pattern reflects transcriptional activation via liganded-ER binding that is described as a cyclic process for some target genes such as cathepsin D [60]. At first, transcription rates of target genes are upregulated and subsequently decline, followed by a repeated increase.

Whereas Taspase1 was hardly detectable in non-induced 293T cells, the rapid estrogen-stimulated upregulation might explain that 293T cells usually express no or low amounts of this protease although the identification of Taspase1 in 2003 relied on an isolation from this cell line. In line with this, we identified an imperfect estrogen response element (ERE) about 7.8 kbp upstream of the *TASP1* promoter (Figure 2B). Notably, it is described that some EREs are located up to 10 kb from transcriptional start site [61] but distal ERα binding sites can be anchored at gene promoters through long-range chromatin interactions [62].

As the estrogen response pathway utilizes Topo II-mediated transient DSBs for the initiation of regulated gene transcription, also the level of phosphorylated histone variant H2AX, a reliable biomarker for DNA double-strand breaks, was investigated. Following estrogen stimulation, γH2AX levels rapidly increased coinciding with Topo II upregulation. Unfortunately, we could not attribute the latter to one specific Topo II isoform as the used antibody recognizes the conserved core domains of both variants. Nevertheless, the DSBs induced after E2 treatment correlate with the increase in Topo II concentration, presumably reflecting the phenomenon of hormone-induced transcriptional activation via Topo IIβ-mediated DSBs.

However, DSBs are also often generated during S-phase, e.g., when a replication fork encounters a DNA lesion. To thus exclude that the appearing DSBs are associated with DNA replication, cells were arrested in G1-phase with lovastatin before E2 treatment. Again, hormone stimulation provokes an increase in DSBs, detectable with a delay of approx. 2 h (Figure 2C and Appendix A). Notably, constant levels of Cyclin E indicate that arrested cells did indeed not progress into S-phase. Upregulation of the early response gene c-myc, which is also estradiol-inducible, is presumably a direct product of gene transcription regulated by hormone-initiated, scheduled DSBs.

In sum, Taspase1 expression was up-regulated within the same timeframe as the aforementioned key players of the cellular estrogen response. As we could also reveal Topo II as a novel interaction partner of Taspase1, it was tempting to speculate that the TASP1 gene is not a mere estrogen target gene, but that Taspase1 might rather occupy a direct function within the process of estrogen-stimulated, Topo IIβ-mediated transcriptional regulation. To further study Taspase1 function in the context of estrogen-driven transcription, we decided to use 293T cells as those cells are embryonic cells with basal expression of ERα. In contrast to high estrogen-responsive breast cancer cells (such as MCF-7) with aberrant estrogen signaling, a more physiological estrogen response is expected in 293T cells.

Thus, we first analyzed whether estrogen treatment might alter Taspase1 complex composition. Therefore, cells were either treated with 50 nM E2 or with DMSO for 1 h before immunoprecipitating Taspase1-GFP. To capture only chromatin-bound Taspase1, the IP was conducted from chromatin fractions instead of whole cell lysates. The analysis of co-isolated proteins revealed a clear increase in Topo II and γH2AX in Taspase1-GFP complexes after E2 stimulation (Figure 3A).

Notably, the association of Taspase1-GFP with chromatin emerged to be independent of hormone stimulation. The increase in Topo II and γH2AX in the chromatin-bound Taspase1 complexes shortly after E2 exposure (Figure 3A) indicates that Taspase1 might be present at chromatin locations directly on-site, where Topo IIβ-mediated DNA cleavage occurs. 

The accumulation of Taspase1′s substrate MLL1 in the immunoprecipitates after hormone exposure (Figure 3A) furthermore implies that also epigenetic modifiers such as MLL can be deposited at such hormone-induced, Topo II-mediated-transcriptional start sites at DSBs. Attachment might be mediated via the hormone receptor binding motif within MLL or by a direct protein-protein interaction with Taspase1, representing an additional binding interface. At the same time, this would enable Taspase1 to cleave its substrate MLL, thereby establishing the well-known H3K4 epigenetic signature. The latter is indeed strongly increased in Taspase1-complexes after hormone treatment. After DNA double-strand breakage followed by subsequent religation, the loosened and thus relaxed chromatin structure allows binding of other transcription factors, possibly including further Taspase1 targets such as TFIIA and USF2. Proteolytic cleavage of these transcriptional regulators by Taspase1 might in turn facilitate to adapt their mode of action to the need of hormone stimuli-invoked genetic programs.

### 3.3. Taspase1 Directly Binds Both Isoforms of Topoisomerase II

As we identified Topo IIα/β as components of immunoprecipiated Taspase1-GFP complexes (Figure 3A) and co-localized with Taspase1 in HeLa cell nuclei and nucleoli (Appendix A), we further investigated the interaction of endogenous Topo II and Taspase1 utilizing the Duolink^®^ PLA technology (Appendix A). Briefly, HeLa cells were fixed and incubated with primary antibodies specific for Topo IIα/β and Taspase1 followed by an incubation with PLA probes, oligonucleotide-modified secondary antibodies. Both PLA probes have to be in close proximity to become covalently joined through enzymatic DNA ligation, assisted by circle-forming connector oligonucleotide. Finally, the generated circularized DNA molecules are amplified by a polymerase, resulting in single-stranded DNA molecules that are visualized as distinct foci by fluorescently labeled complementary oligonucleotide probes. Number and intensity of the foci were evaluated by fluorescence microscopy (Figure 3B and Appendix A). Of note, the negative control incubated only with anti-Taspase1 antibody (but with both PLA probes) showed only low levels of background signal (on average about two PLA foci per nucleus) in contrast to the Topo IIα/β antibody negative control, which resulted in higher background signals (on average approximately nine PLA foci per nucleus) (Figure 3B). Nevertheless, with a median of about 45 foci in the interaction sample (Taspase1 + Topo IIα/β), a significant increase in PLA foci was revealed. As the PLA method detects only molecules in close proximity (<40 nm), we conclude that the binding of Topo II to Taspase1 is presumably relying on a direct protein-protein interaction. However, the antibody used for co-IPs and PLAs recognizes the highly conserved Topo II core enzyme consisting of the ATPase- and the DNA binding domain. As such, we were unfortunately not able to discriminate between the two Topo II isoforms. Therefore, we next performed co-transfection experiments with isoform-specific expression constructs of Topo II. Briefly, HEK 293T cells were transiently co-transfected with plasmids encoding Topo IIα-GFP or Topo IIβ-GFP [49] in combination with wild-type Taspase1-HA coding plasmids and GFP fusion proteins were precipitated from the respective cell lysates as described. Immunoblot analysis revealed that less full-length Topo IIβ-GFP was expressed and immunoprecipitated compared to Topo IIα-GFP. However, in relation to the bait-protein, comparable amounts of Taspase1-HA were co-precipitated with each Topo II isoform (Figure 3C). This strongly argues against isoform-specific binding and suggests that the binding site might be rather located within the conserved Topo II core domain pivotal for DNA binding.

### 3.4. Taspase1 Facilitates the Formation of Topo IIβ-Mediated DNA-Double Strand Breaks

As a direct consequence of our findings, we next aimed at dissecting the exact function of Taspase1 in Topo IIβ-mediated DNA-cleavage. For that purpose, 293T cells expressing GFP, Taspase1-HA, Topo IIα-GFP, Topo IIβ-GFP or Taspase1-HA together with each Topo II isoform were treated with etoposide or DMSO followed by subsequent cell lysis. Etoposide arrests Topo II in the covalent cleavage complex with nucleic acids as it inhibits the DNA religation step. First, we could demonstrate that expression of Taspase1 is sufficient to enhance the formation of DSBs to an extent comparable to Topo IIα, but a little less pronounced compared to Topo IIβ (Figure 4A, left panel). Most importantly, co-expression of Taspase1-HA with both Topo II isoforms results in the most prominent γH2AX-signals. In general, treatment with etoposide leads to an obvious γH2AX increase reflecting unrepaired DNA DSBs upon Topo II poisoning, but still reveals the same trends (Figure 4A, right panel): Taspase1 enhances the amount of DSBs compared to the GFP-control, and co-expression of Topo II, in particular of Topo IIβ with Taspase1 again exerts the strongest effects. Thus, we conclude that Taspase1 promotes the formation of Topo II-mediated DNA DSBs prerequisite for the transcription of estrogen-regulated genes.

To now further discriminate between direct and indirect effects, we utilized a Topo IIβ DNA cleavage assay. Here, the supercoiled plasmid pBR322 is incubated with recombinant Topo IIβ and increasing concentrations of recombinant Taspase1-His. After subsequent protein digestion with Proteinase K, the samples were analyzed by agarose gel electrophoresis. The activity of Topo IIβ generates DSBs and, thus, a linearized or open circle (oc) form of the previously supercoiled (sc) plasmid. As expected, untreated pBR322 primarily exists in its supercoiled form (Figure 4B, lane 2), which is partially linearized in the presence of 4 units of Topo IIβ (Figure 4B, lane 3). With increasing concentrations of recombinant Taspase1-His, the amount of supercoiled plasmid declines in favor of more prominent linearized and open circle species (Figure 4B, lanes 4–8). This suggests that Taspase1 might directly facilitate the formation of Topo IIβ-mediated DNA DSBs.

### 3.5. Taspase1 Is a DNA-Binding Protein

Based on those findings, we next investigated if Taspase1 itself might be capable of direct DNA binding. Indeed, electrophoretic mobility-shift assays (EMSA) could characterize Taspase1 as a DNA-binding protein (Figure 4C). Addition of increasing amounts recombinant Taspase1-His induced gradual shifts of supercoiled plasmid pBR322 (Figure 4C, lanes 3–7). Notably, a maximal size shift was reached with protein concentration above 2.5 µM (Figure 4C, lane 7). Thus, we conclude that the plasmid DNA might be saturated with bound Taspase1-His corresponding to a molar ratio of 1:140. 

In sum, the reduced electrophoretic mobility of pBR322 upon gradual addition of increasing amounts of recombinant Taspase1-His suggests the formation of a complex by presumably direct interaction with the plasmid DNA. This now offers a mechanistic explanation how Taspase1 might facilitate Topo IIβ-mediated DNA DSBs.

### 3.6. Taspase1 Is a Multifunction Co-Activator of Estrogen-Driven Transcription

Finally, we aimed to reveal whether the transcriptional estrogen response that utilizes Topo IIβ-mediated DNA DSBs is indeed Taspase1-dependent. Therefore, Taspase1 was downregulated with small interfering RNAs (siRNAs) for 24 h or 48 h (Figure 5A). A non-targeting siRNA was used as a control for unspecific knockdown effects, and GAPDH served as loading control. Notably, the expression of the estrogen-stimulated gene c-myc was markedly reduced upon Taspase1 downregulation, in particular after 48 h. Moreover, trimethylation of histone H3 lysine 4 (H3K4me3) decreases upon Taspase1 silencing. This is in perfect agreement with the estrogen-induction time courses (Figure 2A) where Taspase1 expression correlates with the degree of H3K4 trimethylation (Figure 5A). Together, these results indicate that Taspase1 might enable H3K4me3 epigenetic labelling via cleavage-triggered MLL activation in an estrogen-dependent manner. Hence, it can be concluded that Taspase1 is also responsible for the necessary epigenetic signature that allows the maintenance of hormone-activated transcriptional programs. Moreover, our study suggests for the first time a pivotal role of Taspase1 in the cellular estrogen-pathway and classifies Taspase1 as a multifunctional co-activator of estrogen-driven transcription.

## 4. Discussion

A plethora of pivotal biological processes is triggered by the steroid hormone estrogen, in particular cell proliferation and differentiation. Central participants in the classical estrogen signaling pathway are the nuclear receptors ERα and ERβ, responsible for hormone binding and forwarding the signal to the nucleus. Here, the ligand-bound receptors subsequently stimulate or inhibit transcription of target genes via binding to estrogen response elements (EREs), sequences close to or within promoters of those genes or even located up to 10 kb from their transcriptional start site [61]. However, ERs imperatively require co-regulators that fulfil important functions for example in chromatin remodeling or in transcription initiation, finally contributing to the activation of target genes.

Here, we now identified Taspase1 as such a crucial co-regulator of estrogen-driven transcription. Importantly, we could demonstrate *TASP1* transcriptional induction by estrogen and also detect a potential ERE within the regulatory region of the *TASP1* gene. Although bioinformatic tools such as the CiiiDER software [63] predict ERα binding to this site, experimental evidence by ChIP seq data is still lacking. In addition, the identification of novel interaction partners such as Nucleolin and Topo II could reinforce the functional connection with the cellular steroid hormone response pathway. Nucleolin as well as Nucleophosmin1, which were already assigned as members of the Taspase1 interactome, are both components of a co-repressor complex [64] which is, e.g., associated with the estrogen-inducible promoter of the pS2 gene during the uninduced state, thereby blocking transcription of this classical model estrogen target gene [65]. For the transcriptional activation of hormone-inducible genes, Topo IIβ function has proven to be absolutely pivotal. In detail, the enzyme generates a transient DNA double-strand break (DSB) within the respective promoter, leads another DNA segment through the cut and, lastly, religates the break. This Topo IIβ-catalyzed reaction resolves topological barriers such as DNA supercoils or knots in the regulatory region of target genes, thereby opening the chromatin and allowing the transcription machinery to gain rapid and unhindered access. In contrast to the sporadic endogenous DNA damage which can occur as a harmful byproduct of replication and transcription, such hormone-induced DSBs in promoter regions are scheduled events which are absolutely essential for the activation of certain transcriptional programs [66]. Our study has now unraveled the protease Taspase1 as a novel interactor of this key player of estrogen-driven transcription. We could furthermore characterize this interaction as transient and estrogen dependent. Moreover, the association of Taspase1 with Topo II was also verified in a cellular context in PLA studies, an approach with preserved native conditions and detectable co-localizations below 40 nm, thereby indicating a presumably direct protein-protein interaction.

Our data further indicate that Taspase1 directly facilitates transient Topo IIβ-catalyzed DNA double-strand breaks in a dose-dependent manner. Interestingly, this novel function of Taspase1 is independent of its proteolytic activity but can rather be attributed to direct DNA binding which was also revealed in this study (Figure 4C). As Taspase1 was first identified as a protease, further supplemental, independent functions may have been overlooked so far. But nowadays, the concept of “one enzyme—one function“ is obsolete since many multifunctional proteins have been identified and characterized in the last decades [67]. Moreover, one fundamental new insight of our study is the fact that the catalytic and DNA-/protein-binding activities of Taspase1 might be largely independent from one another as the proteolytically inactive mutant is also able to bind Topoisomerase II, γH2AX and Nucleolin with comparable efficiencies. Although Taspase1 does not harbor a classical DNA binding domain, it should be considered that the basic amino acid residues of nuclear localization signals (NLS) are sometimes also involved in DNA binding. For several transcription factors such as MyoD or NF-κB, the NLS is an elementary part within the DNA binding domain [68]. Indeed, the bipartite NLS of Taspase1 is located in its flexible, surface-exposed loop region of Taspase1, which renders the interaction of this cluster of basic residues to nucleic acids possible. On the other hand, it is still not completely understood how Topo II selects its specific cleavage sites on the DNA. Currently, the general consensus is that specific nucleotide sequence-based recognition sites might not serve as the main defining factor. Physiologically, this makes perfect sense as the cleavage of inducible genes even in absence of a stimulus should not be permanently allowed. Maybe Taspase1 operates as an adapter protein between Topo II and destined DNA target sequences within the regulatory regions of estrogen-inducible genes. As Taspase1 itself is rapidly upregulated by estrogen, a self-reinforcing process might govern subsequent E2-mediated transcriptional adaption: low basal Taspase1 protein levels mediate initial Topo IIβ-triggered DNA DSBs leading to transcriptional activation of E2-responsive genes, including the *TASP1* gene itself. This feedback loop would mechanistically result in a prompt signal amplification. Indeed, Taspase1 knock down studies support this idea as they revealed a subsequent downregulation of E2-responsive genes such as c-myc and c-fos. Nevertheless, further studies applying e.g., ChIP sequencing, would be of utmost importance to elucidate which conserved or functional DNA motifs are preferentially bound by Taspase1.

Finally, we envisage the following scenario: Taspase1 resides at regulatory chromatin regions of hormone-inducible genes. Here, together with Nucleolin and NPM1, it binds to the promoter-associated repressor complex (Figure 5B). After E2 stimulation followed by DNA-binding of the respective hormone ligand-receptor complex and initial chromatin rearrangements, Taspase1 makes contact with Topo II and recruits the enzyme to DNA sites destined for breakage-triggered activation. As Taspase1 also interacts with the NPM1/Nucleolin/PARP1 repressor complex, it is conceivable that it facilitates the switch of this complex, thereby making way for Topo II. As such, Taspase1-supported Topo IIβ-mediated DNA DSBs finally allow an open chromatin conformation and enable the transcription machinery to gain rapid access for the immediate synthesis of mRNA. This resembles another situation where an auxiliary protein is described to regulate the interplay of Topo II with chromatin: Dykhuizen et al. (2013) demonstrated that chromatin remodeling BAF (SWI/SNF) complexes directly interact with Topo IIα and facilitate its binding to DNA via BRG1 (also known as SMARCA4)-dependent chromatin remodeling [69]. Notably, BRG1 is also able to recruit Topo I to chromatin [70]. Analogously, BRG1 is part of the ATP-dependent chromatin remodeling complex SWI/SNF, whereas Taspase1 can associate with the MLL complex which is also involved in the modification of chromatin architecture.

As such, our work comprehensively demonstrates that Taspase1 assists Topo IIβ in the generation of physiologically relevant, estrogen-induced transient DNA DSBs within estradiol-responsive promoters. Moreover, our results unravel that Taspase1′s proteolytic activity contributes to the estrogen signaling pathway: nuclear steroid hormone receptors are often accompanied by epigenetic writers and modifiers to perpetuate altered gene expression patterns. MLL is such a chromatin-modifying enzyme that also plays a role in hormone-induced transcription. Certain cell states that are triggered by stimulus-induced transcriptional programs have to be maintained for a longer period of time or even to persist permanently. This requires, e.g., epigenetic labelling to manifest gene activation [71]. This as well as other studies show that the MLL1 protein and its family members are also involved in estrogen-driven transcription. MLL’s C-terminal SET domain harbors an intrinsic histone methyltransferase activity which mediates trimethylation of H3K4 (H3K4me3), a specific signature for epigenetic transcriptional activation [13]. Proteolytic processing of this enzyme by Taspase1 is indispensable for a proper histone H3 methyltransferase activity of MLL [12]. E2 treatment results in an increased H3K4 signature (Figure 2A) whereas Taspase1 silencing reduces this epigenetic mark (Figure 5A). This, in turn, renders Taspase1-mediated cleavage responsible for the pivotal MLL activation and the epigenetic activation of estrogen-stimulated genes.

MLL, however, is not the only Taspase1 substrate that can enter the scene of estrogen stimulated transcriptional initiation. Additionally, the general transcription factor TFIIA that is part of the transcription preinitiation complex initiating mRNA synthesis, is proteolytically processed by the protease [72]. Uncleaved TFIIA enables transcription of the negative cell cycle regulators p16 and p19, in turn blocking cell cycle progression. Taspase1 cleavage modulates TFIIA activity resulting in the initiation of an adapted genetic program with enhanced cell proliferation [58,72]. Certainly, such a regulation of TFIIA activity requires a certain stimulus. As TFIIA also features an estrogen-dependent expression it is likely that the initiation of this proteolytically triggered functional fine-tuning might be caused by hormone stimulus. In general, a distinct function in cell cycle progression has been ascribed to Taspase1: Taspase1^−/−^ mouse embryonic fibroblasts only poorly proliferate because of a disrupted cell cycle. They revealed a profound downregulation of cyclins E, A and B and an increased expression of p16, indicating a cell cycle block at G1/S transition [12]. It remains to be investigated if Taspase1′s newly identified role in the cellular estrogen response is also directly linked to the cell cycle. A cell progression through G1-phase and G1/S transition is a critical step influenced by a plethora of external signals. Before the cell reaches the restriction point, external signals, e.g., growth factors and hormones, are integrated and assessed to decide if the cell can enter S-phase. Certainly, the cascades which control the passage of the G1 restriction point are largely depending on cell type and context. However, in estrogen receptor-positive cells estradiol promotes proliferation by facilitating the G1/S transition [73,74]. As such, it is tempting to speculate that the E2-induced transcription program, which is mediated (amongst others) by Topo II, Taspase1, Topo II-generated DSBs and MLL, might also trigger the G1/S transition. Finally, it remains to be investigated if Taspase1 also co-activates transcription triggered by further stimuli such as androgen, glucocorticoids, progesterone etc.

Last but not least, our data may give an explanation for Taspase1′s overexpression in leukemia where chromosomal translocations are quite common. We have learned that Taspase1 facilitates Topo II-dependent DNA DSBs upon hormone stimulation. Usually, such DSBs are either religated by Topo II itself or, alternatively, are quickly sensed and repaired by the cell’s DNA damage response. However, in case of a malfunctioning estrogen-signaling cascade, there is an increased risk of unrepaired or incorrectly repaired DSBs, leading to programmed cell death or chromosomal translocations. Imagine the following scenario: Genes that are activated at the same time, e.g., as response to a certain stimulus, are transcribed in parallel and in close proximity to each other in so called transcription factories [75,76]. If Topo II-catalyzed DSBs are now generated within regulatory gene regions, instead of the fragments from one gene being ligated, a cross-over with an adjacent, also “broken” gene could occur. Consequently, this would result in the assembly and ligation of two fragments from different genes on different chromosomes, depicting the phenomenon of chromosomal translocations. Moreover, Topo II poisons, which are indeed widely spread in cancer therapy, exploit the hazardous potential of Topo II-generated cytotoxic DNA damage, as unrepaired DSBs ideally lead to programmed death of cancer cells. However, following chemotherapy with the Topo II inhibitor etoposide, harmful therapy-related, secondary leukemia arises in 2–15% of cancer patients, harboring chromosomal rearrangements [77,78,79,80]. This underlines once more that although Taspase1-facilitated transient DSBs catalyzed by Topo II have a high physiological relevance as an initial event for stimulus-provoked transcription, those scheduled DNA breaks also exert a great hazard potential if serious disturbances occur in this pivotal signaling cascade. First identified as the protease responsible for processing MLL and thus for leukemia development [2], Taspase1′s relevance for various solid cancer types is now widely accepted [1,11]. Here, it functions as a non-oncogene addiction protease orchestrating cancer cell proliferation and apoptosis. Although its molecular pathobiology is still not completely understood, overexpression of Taspase1 in tumors mostly correlates with poor patient survival, again underlining its relevance for solid tumor entities, e.g., including epithelial tumors of the head and neck (HNSC) [81]. Interestingly, besides “classical” hormone-dependent cancers including those of the breast or prostate also HNSC might depend on regulation by steroids such as estrogen and androgen and their receptors [82,83,84]. Additionally, first-line tumor therapy primarily relies on producing excessive DNA damage leading to direct cell death, such as induced by irradiation or DNA-intercalating agents, e.g., applied in but clearly not limited to HNSC. As such, our data suggest to the field to further investigate the relevance of these novel findings and thus of the Taspase1/ER-axis for various malignancies employing tumor-specific in vitro and in vivo models.

## 5. Conclusions

In sum, our study provides a much clearer picture now: Taspase1 can no longer be regarded as a protease that cleaves functionally unconnected substrate proteins. Its sphere of action is rather embedded in a specific signaling cascade, the cellular estrogen pathway, thus classifying Taspase1 as a multifunctional co-activator of estrogen-driven transcription. At its operation site—the chromatin—Taspase1 contributes to hormone signal forwarding by several means:Taspase1 facilitates estrogen-induced Topo IIβ-mediated transient DNA double-strand breaks that are required for the initiation of stimulus-triggered transcription.Taspase1 enables H3K4me3 epigenetic labelling via cleavage-triggered activation of MLL.Taspase1 cleaves further transcription factors and other proteins to adapt their function to a changing cellular environment.

## Figures and Tables

**Figure 1 cells-12-00363-f001:**
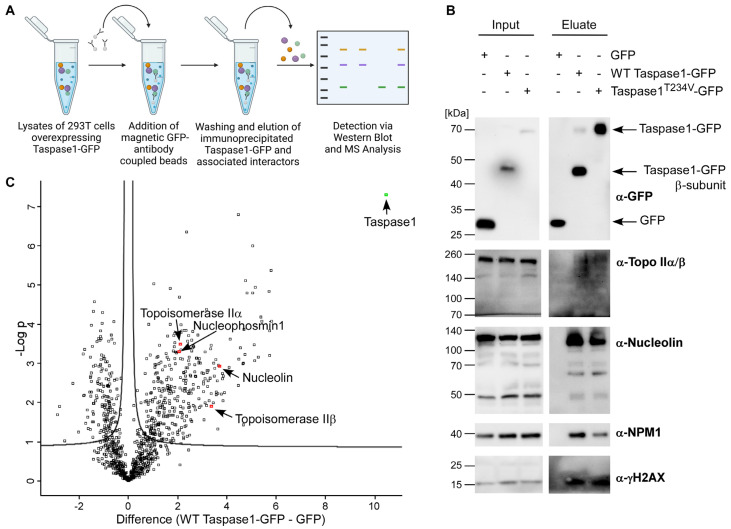
Taspase1 participates in the protein network that mediates estrogen-driven transcription. (**A**) Workflow of Taspase1-GFP co-immunoprecipitation (co-IP) followed by immunoblot analysis or mass spectrometry (MS). This illustration was created with BioRender.com. (**B**) Identification of novel Taspase1 interactors by co-IP combined with immunoblot analysis. (**C**) MS analysis of Taspase1-GFP co-immunoprecipitates. The enrichment of identified proteins in the Taspase1-GFP eluate (x-axis, intensity WT Taspase1-GFP–GFP) is plotted against the −log10-transformed *p*-values (y-axis), which is a measure for the reproducibility of the respective protein enrichment. The bait protein Taspase1-GFP is indicated in green. Further enriched proteins in the upper right section represent interactors of Taspase1. Topo II is among the most-enriched proteins in the Taspase1-GFP immunoprecipitates, as well as known interactors of Taspase1 such as NPM1.

**Figure 2 cells-12-00363-f002:**
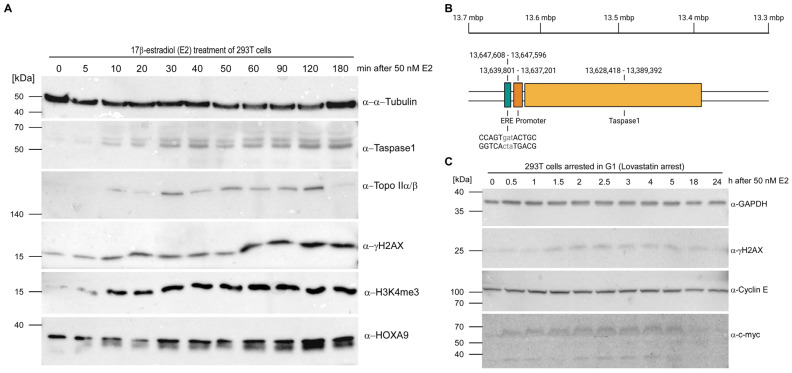
*TASP1* is an estrogen-responsive gene. (**A**) Programmed DNA DSBs appear shortly after treatment of 293T cells with E2. Topo II and Taspase1 are synchronously upregulated as detected by immunoblot. (**B**) Visual representation of the identified Estrogen Response Element (ERE) located 7.8 kbp upstream of the *TASP1* promoter. *TASP1* gene and flanking region up to −10 kb were screened for a plethora of different EREs. This illustration was created with BioRender.com. (**C**) E2 treatment of G1-arrested cells also provokes an increase in DSBs indicating that the emerging DSBs reflect the phenomenon of hormone-induced transcriptional activation via Topo IIβ-mediated DSBs instead of replication-associated DNA lesions.

**Figure 3 cells-12-00363-f003:**
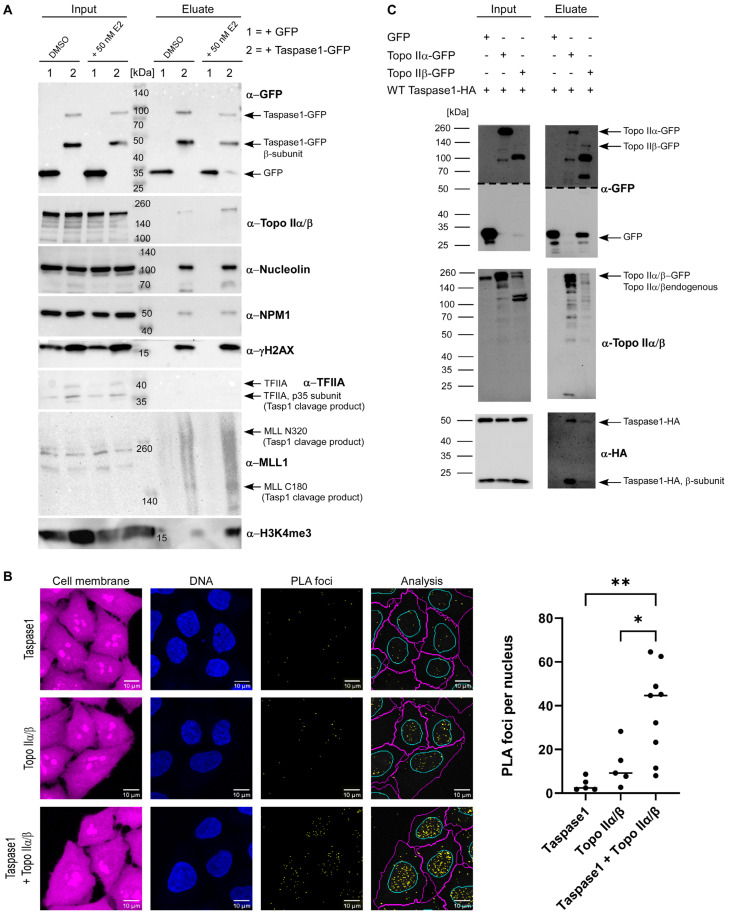
Topo IIα/β interacts with Taspase1. (**A**) Analysis of the Taspase1 complex composition after hormone stimulation. HEK 293T cells were transiently transfected with the indicated GFP-fusion constructs. After 24 h, cells were either treated with 50 nm E2 or mock-treated for 1 h. Chromatin fraction (Input) and immunoprecipitates (Eluate) were separated by SDS-PAGE and analyzed by immunoblot. (**B**) PLA foci analysis of Taspase1 and Topo IIα/β within the nucleus of HeLa cells. Distinct Taspase1-Topo II foci reflect a close co-localization. Nuclei were stained with Hoechst33342. Scale bars, 10 µm. Quantification of PLA signals per nucleus and representative images are shown. Results and median of independent experiments (*n* ≥ 5) each with 150–450 cells. Asterisks indicate significant differences evaluated by one-way ANOVA followed by Tukey’s multiple comparison test (*: *p* ≤ 0.05, **: *p* ≤ 0.01). (**C**) Topo IIα-GFP and Topo IIβ-GFP were precipitated from 293T cell lysates. Ectopically expressed Taspase1-HA is co-precipitated with both Topo II variants demonstrating that binding is not isoform-specific.

**Figure 4 cells-12-00363-f004:**
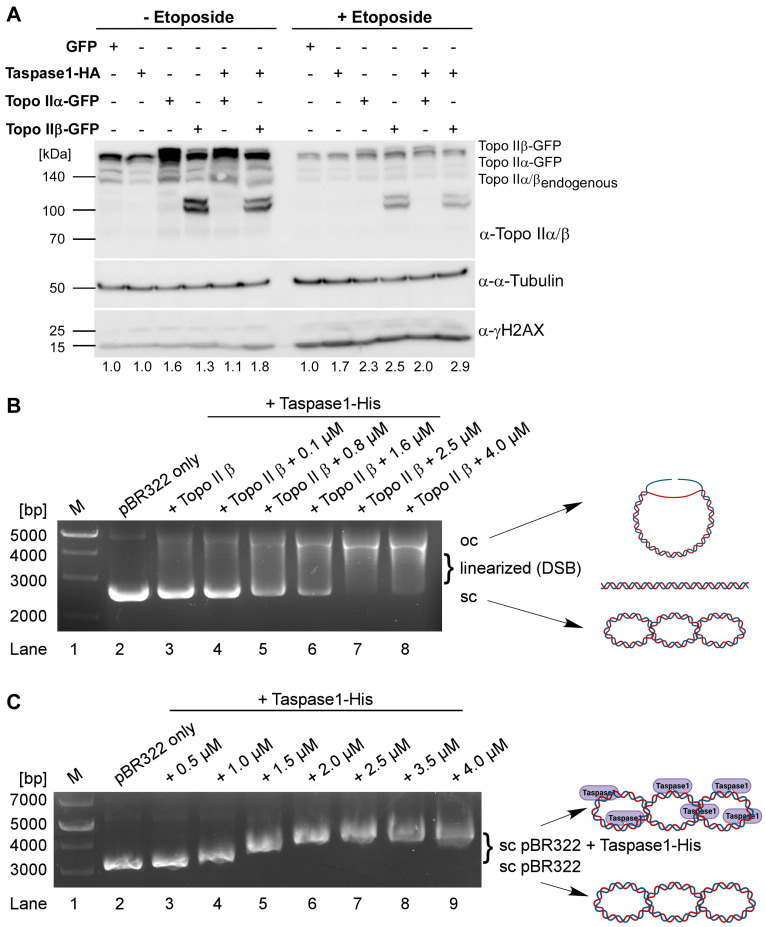
Taspase1 facilitates Topo IIβ-mediated DNA-double strand breaks. (**A**) Analysis of DNA DSB formation dependent on Taspase1 and Topo II expression. The 293T cells expressing GFP, Taspase1-HA, Topo IIα-GFP, Topo IIβ-GFP or Taspase1-HA together with the indicated Topo II isoforms were treated with etoposide or DMSO, and cellular γH2AX levels were assessed as a readout for DNA DSBs. (**B**) Taspase1 is sufficient to enhance Topo IIβ-cleavage of supercoiled pBR322 plasmid DNA. pBR322 plasmid was incubated in the absence (lane 2) or presence of recombinant Topo IIβ (lanes 3–8) and increasing concentrations of Taspase1-His. Plasmid illustrations were created with BioRender.com. (**C**) EMSA gel-shift assays reveal a retarded electrophoretic mobility of Taspase1-bound pBR322 plasmid DNA. Agarose gels were stained with HD-green. Plasmid illustrations were created with BioRender.com.

**Figure 5 cells-12-00363-f005:**
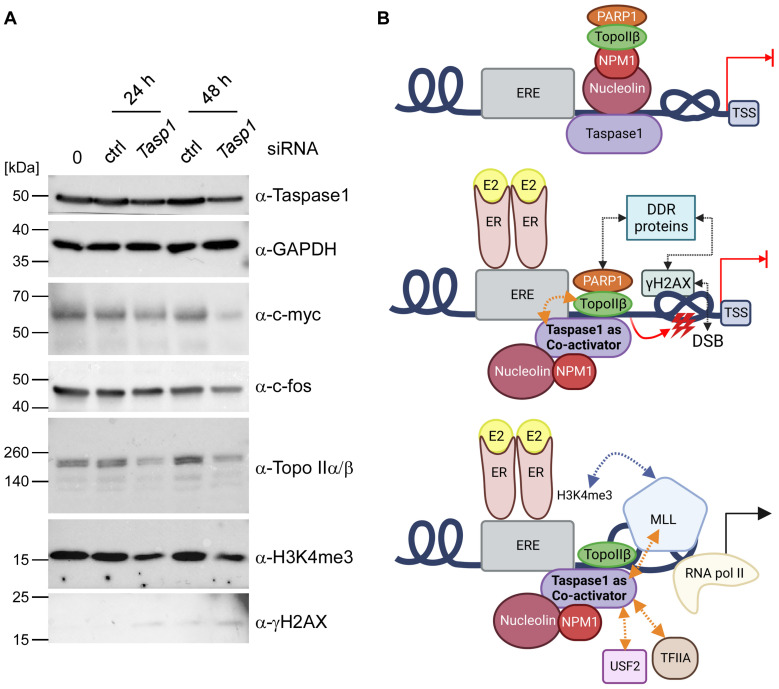
Taspase1 is a co-activator of estrogen-driven transcription. (**A**) Downregulation of Taspase1 affects the transcriptional estrogen response. Western Blot analysis of whole cell lysates of untransfected 293T cells or 293T cells 24 h and 48 h after transfection with Taspase1 siRNA or control siRNA unravels a partial Taspase1 knock-out coupled with an attenuated expression of estrogene-responsive genes. (**B**) Model of Taspase1-influenced estrogen-driven transcription: In an uninduced state, the co-repressor complex (Nucleolin, NPM1, PARP1 and Topo IIβ) is bound to promoter regions of estrogen target genes, thereby preventing transcriptional initiation. In addition, Taspase1 is associated with this complex via Nucleolin and NPM1. Estrogen-stimulation leads to a spatial rearrangement of the co-repressor complex and enables Taspase1 to bind and successively promote Topo IIβ to induce transient DNA double strand breaks (DSBs), labeled by γH2AX which is presumably associated with the complex via DNA damage response (DDR) proteins. Topological barriers such as DNA knots or supercoiling are solved, allowing unhindered access for the transcription machinery. Moreover, Taspase1 cleaves different substrates, e.g., TFIIA and USF2 to adapt their function to changing physiological conditions. Furthermore, Taspase1 enables H3K4me3 epigenetic labelling via cleavage-triggered MLL activation in an estrogen-dependent manner. This epigenetic signature may thus allow to maintain distinct hormone-activated transcriptional programs. This illustration was created with BioRender.com.

## Data Availability

The mass spectrometry proteomics data for the on-bead digestions have been deposited to the ProteomeXchange Consortium via the PRIDE [58] partner repository (https://www.ebi.ac.uk/pride/archive/) with the dataset identifier PXD035546.

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
