# Peer review of "Taspase1 Facilitates Topoisomerase IIβ-Mediated DNA Double-Strand Breaks Driving Estrogen-Induced Transcription"

_cells, 2023, doi:10.3390/cells12030363_

Round 1
Reviewer 1 Report
The Authors investigate Taspase1, a protease with unclear biological function. They present data supporting their novel hypothesis that Taspase1 is a mediator of transcription induction by estrogens involving two Taspase1 activities (i)DNA binding and Topoisomerase II recruitment at target gene promoters, causing transient DNA double strand breaks to release chromatin blocks (ii) cleavage of MLL1, thus activating its histone methylase function yielding H3K4me3, an open chromatin component
The manuscript is very well written with background, hypothesis , experimental set up and conclusions clearly laid out for each experiment. The Introduction provides all the necessary background. The Discussion cleverly develops the new model based both on the present data and on publications of other groups, and clearly schematized in Fig 5.
Comments
1. A key point in the study is the presence of DNA double strand breaks (DSB) associated to Taspase1 recruitment of Topoisomerase II (Topo II). As stated on line 511 the read out of DSB presence is the expression of gamma H2AX. Here the Authors have identified gamma H2AX among Taspase1 interacting proteins. In order to clearly demonstrate the ability of Taspase 1 to recruit Topo II and lead to local DSB occurence would it be possible to show co-localisation of gamma H2AX and Taspase1 by Duolink PLA? A putative issue is that since the DNA ends generated by Topo II cleavage are not released from the enzyme, could gamma H2AX bind to Topo II-mediated DSB? Maybe gammaH2AX interaction with Taspase1 occurs at a distance from Topo II since it is known gammaH2AX can bind up to mega base distances from a DSB.
2. Fullwood et al Nature 2009 have shown that distal ERα binding sites were anchored at gene promoters through long-range chromatin interactions, suggesting that ERα brings its target genes together by chromatin looping thus allowing for their coordinated transcriptional regulation. The ERE the authors have identified in silico in the Taspase1 upstream sequences is not a "bona fide" ERE in contrast to what is mentioned line 574, because its ability to bind the ERE and to activate Taspase1 gene expression has not been demonstrated. An evidence in favor of actual ERE alpha binding to this site in different cell types could be found by searches in the datasets of the ENCODE project or in the Fullwood et al 2009 ChIP seq study or other ones.
3. The intensity changes in immunoblot bands are often difficult to evaluate: the authors should perform densitometry scanning for the blots related to estrogen induction (Fig 2) and gamma H2AX in Fig4A
Minor comments
Line 52: the gene (italics)/protein nomenclature should be adapted to the species mentioned in the whole manuscript : for human (all capital letters): HOX gene, HOX protein, for mouse: Hox gene, Hox protein
Lines 101 and 102: change "culturing" to "culture"
Line 113: change "without previous Lovastatin release" to "in Lovastatin containing medium"
Line 114: . typo in 17 beta estradiol
Line 115: in in
Line 122: does the protease inhibition cocktail used also affect Taspase1 activity?
Line 123 : change "two times" to "twice"
Line 127: Tubulin was used as a loading control, not its antibody!
Lines 131/132 and lines 136/137: in standard TC dish 100
Line 140/141: The siRNA single strands were pooled
Line 167: what is the % acrylamide used for SDS-PAGE?
Line 178: For visualization of immune complexes
Line 204: "foci" is the plural of "focus", a latin word, so no need to add s!
Line 206: words missing at the end of the sentence "whole cell"?
Line 213: was purified on (not one) a HisTrap...
Line 222: mention that pBR322 is supercoiled
Line 243: Surfactant
Line 262: who is "Dr Maisch" ?
Line 345: S1 Table 1 was not available for my review
Line 360: Change " is estradiol..." to "was estradiol... "
Line 431: Change " adopt" to "adapt"
Fig3A: the MLL1 cleavage products are very difficult to observe in the smear of that immunoblot, could the authors provide a cleaner one?
Line 531: consequentially
regions Line 537: change "estrogen time course" to " estrogen induction time course"
Fig 5A: it is not clear what happens to gamma H2AX levels upon Taspase1 siRNA addition, it seems the immunoblot signal intensity increases with Taspase 1 knock down! Could a densitometry or another experiment clarify the matter?
Fig 5 B: Model of Taspase 1 influenced estrogen-driven transcription: PARP1 was not mentioned among the proteins associated with the Taspase1-GFP complex in Fig1, why is it shown here? In contrast gammaH2AX was found among Taspase1 interactors, why is it not shown here?
Line 562: could the Authors comment on putative Taspase1 functions related to its associated proteins linked to the nucleolus, RNA Pol III transcription and ribosomes?
Line 568: "sequence close to or within promoters", this is not correct since many EREs have been identified in more distant regions (as is the case for the Taspase1 putative ERE identified in silico in the present study) acting like enhancers
Line 594: the expression "in vivo" is normally referring to live animals, so should be avoided here, change to "in vivo approach with preserved native..."
Line 623/624: the word "depletion" is not appropriate here since the siRNA only caused a limited decrease in Taspase1 ( Fig 5A upper line), the Authors might use " knock down" instead
Line 677: 's is only used for persons, here "A cell progression" is grammatically correct
Line 697/698: correct English in the sentence "not ... but"
Line 739: "geb" is not English
Line 741: capital letter to Topo II
Reviewer 2 Report
Comments: This is an interesting and well-written manuscript on exploring a second functional role of the protease Taspase1 in estrogen-driven transcription. I do have a few questions for the authors to address. First, I am always concerned about alteration of binding properties with GFP tagged expression proteins. Do the authors have any data showing similar co-precipitation blots for Nucleolin, NPM1, a-gH2AX TopoIIa, etc with untagged Taspase1? It would be interesting to compare the stained protein binding profiles of untagged Taspase1 and Taspase1-GFP. Second, it seems like estrogen responsiveness would have far better other cell types than 293T cells. The imperfect palindromic sequences (ln 375) could have been improved upon by using an ERE reporter vector. Since estrogen responsive genes like c-myc are cell-specific and context-specific (ln 680-682) for needed nuclear regulatory proteins in the Taspase-1 complex (e.g. Fig 5), I wonder how widely applicable this new transcriptional role could be for Taspase1 for different cells types which are not estrogen responsive. Third, it would be helpful to explain the expected kinetics of the estrogen-dependent transcription with continued exposure to estrogen since authors describe in the Discussion (ln 590-592) that Taspase-1 interactions are transient and estrogen-dependent. Addressing these comments would be helpful to the reader in better understanding the new role proposed for Taspase-1 given the limitations of the experimental expression systems used and the dependency of cell/context specific nature of estrogen-responsive tissues.
Editorial remarks:
Ln 94: should be “Following this line of reasoning, we could now hypothesize a higher level function of Taspase1…”
Ln 123: should be “scraped” not scrapped.
Ln 133: define WT as (wild type).
Ln 326: please provide citations for USF2 and TFIIA as Taspase1 target proteins.
Figure 1, panel B on a-Topo IIa/b eluate blot. Is there a better blot than this? Unclear what bands are being shown or are not being shown.
